# An epidemiological study of gastrointestinal nematode and *Eimeria* coccidia infections in different populations of Kazakh sheep

**Xiaofei Yan**[1,2], **Mingjun Liu**[3]*, **Sangang He**[3], **Ting Tong**[2], **Yiyong Liu**[4], **Keqi Ding**[5], **Haifeng Deng**[6], **Peiming Wang**[6]

**1** College of Animal Science and Technology, Shihezi University, Shihezi, Xinjiang Province, China, **2** College of Science and Technology, Xinjiang Agricultural University, Urumqi, Xinjiang Province, China, **3** Institute of Biotechnology, Xinjiang Academy of Animal Science, Key Laboratory of Genetic Breeding and Reproduction of Herbivorous Livestock of Ministry of Agriculture, Xinjiang Key Laboratory of Animal Biotechnology, Urumqi, Xinjiang Province, China, **4** Animal Husbandry Terminus, Ili Kazakh Autonomous Prefecture, Yining, Xinjiang Province, China, **5** Animal Husbandry and Veterinary Workstation, Nilak County, Xinjiang Province, China, **6** Zhaosu Horse Farm, Ili Kazakh Autonomous Prefecture, Zhaosu, Xinjiang Province, China

* xjlmj2004@yahoo.com

**Data Availability Statement:** All relevant data are within the paper and its Supporting Information files.

## Abstract

This is an epidemiological study on the gastrointestinal nematode (GIN) and *Eimeria* coccidia infections in Kazakh sheep and the F1 and F2 generations of Kazakh × Texel sheep crosses. A total of 7599 sheep fecal samples were collected from the Zhaosu County and Nilka County in Ili Kazakh Autonomous Prefecture in the four seasons-spring, summer, autumn, and winter of 2019. The parasite causing the infection was identified by the saturated saline floating method, and the infection intensity was calculated by the modified McMaster method. SPSS19.0 was used to evaluate the differences in the fecal egg count (FEC) of for GIN and the fecal oocyst count (FOC) value of for coccidia per sample. The results showed that there were nine types of sheep GIN infections and *Eimeria* coccidia in these two counties of Ililocations, with the dominant parasite species of *Haemonchus contortus*, *Trichostrongylus* spp., and *Ostertagia* spp as the predominant parasites in the sheep. Most of the GIN and coccidia infections in these two regions were mild and moderate. The mean log (FEC) of GIN infection in the Zhaosu area was significantly higher than that in the Nilka area, whereas the mean log (FOC) of coccidia infection in Zhaosu was significantly lower than that of Nilka. The mean log (FEC) of GIN infection in the four seasons was the highest in spring, followed by in summer, then in autumn, and the lowest in winter. The mean log (FOC) of coccidia infection was the highest in spring, followed by in autumn, and was the lowest in summer and winter. The mean log (FEC) of GIN infection and log (FOC) of coccidia infection of Kazakh sheep was significantly higher than the F1 generation, which was then significantly higher than the F2 generation of summer. A positive correlation was found between the EPG and OPG levels in the sheep. These results showed that the GIN and coccidia infection intensities of the F1 generation sheep of Kazakh ×Texel crosses were significantly lower than that of Kazakh sheep paving the way for marker-based resistance selection.

**Funding:** This study was financially supported by the KeyLab funding of Xinjiang Ugrus Autonomous region:"identification of genes with resistance to sheep gastrointestinal nemotode infection by genome-wide association study", The Tianshan Innovation Team (2018D14004) and Scientific Research Project of University of Xinjiang Autonomous Region (XJED2020Y049).

**Competing interests:** The authors have declared that no competing interests exist.

## Introduction

Infection by more than one species of gastrointestinal parasites in sheep while grazing is common. The similarity in infection approach and release patterns of gastrointestinal nematode (GIN) eggs and *Eimeria* coccidia oocysts impair the immune system and reduce the white blood cell count following GIN infection. Either concurrent or successive infection by coccidia then results in the common mixed infections of GIN and coccidia in grazing sheep. This is one of the most serious constraints challenging the production of grazing sheep. Both these infections display similar clinical symptoms ranging mainly from digestive tract inflammation, damage to the integrity of gastrointestinal tissues, to nutritional disorders. These result in gradual weight loss, anemia, limited growth and development, slowed weight gain, reduced efficiency of feed utilization, and exhaustion and even death in sheep in severe cases [1–3]. It adversely affects the quality of the skin, fur, meat, and milk production while increasing the breeding costs and decreasing the economic benefits. This also is one of the key reasons for spring fatigue and thinness that it seriously restricts the development of sheep breeding. Research from many countries such as Brazil [4], Poland [5], Kenya [6], Colombia [7], India [8], and China [9], etc. have reported that mixed infections of GIN and coccidia are common. Studies have also shown a positive correlation between the infection intensity of GIN with that of the infection intensity of coccidia [6, 10]. Both processes viz. the development of the third-stage infective larvae (L3) of GIN and the sporulation of coccidia oocysts require optimal temperature and humidity. Both humidity and temperature play a vital role in the epidemiology of gastrointestinal parasites. Various studies have demonstrated that the increase in the counts of eggs and oocysts was closely related to the local rainy season [8, 11, 12]. For example, in a humid environment, the L3 larvae of *Haemonchus contortus* and other nematodes could reach the infection stage within 46 days to overall increase the fecal egg count (FEC) in the wet season compared to the dry season [11].

It is estimated that the global cost of parasite treatment is as high as tens of billions of dollars each year [13]. While the past few decades have seen the use of anthelmintics for chemical control, more and more nematodes have developed anthelmintic resistance. A study in New Zealand showed that GINs in 43% of farms were resistant to Fenbendazole while GINs in 33% of farms were resistant to Ivermectin at half the dose [14]. As of now globally, three major sheep nematodes viz. *Haemonchus contortus*, *Teladorsagia* (*Ostertagia*) *circumcincta*, and *Trichostrongylus colubriformis* have been reported to be resistant to all major anthelmintics [15]. Reports have shown that 70% of poultry strains were resistant to several anticoccidial medications [16, 17]. The FDA has approved only Decoquinate and Monensin for controlling coccidiosis in poultry, no medications have been approved yet to treat coccidiosis in sheep. In addition, in the United States, anticoccidial medications are not allowed in certified organic poultry production [18]. There is an urgent need for sustainable alternative measures to prevent GIN infection to address the issues of residual drugs and drug resistance. The supplementary approach of choosing sheep that are genetically resistant to gastrointestinal parasites has attracted more attention from researchers. A large number of studies have shown that the resistance to parasites varies among sheep varieties. For instance, Red Maasai [19], Garole [20], Gulf Coast Native [21], Rhön [22] and Barbados Black Belly [23] have been found to have high GIN resistance. Similarly, genetic variations within breeds have also been reported in various sheep populations, including Merino [24], Romney [25], Scottish Blackface [26], and Soay [27], providing a possibility for breeding varieties that are resistant. As the infection intensity, fecal egg count (FEC) and coccidia fecal oocyst count (FOC) predominantly influence the outcome of gastrointestinal parasite infection in sheep, these factors have been used to effectively select resistant sheep varieties [28–30].

This study was aimed at further exploring the difference in GIN and coccidia infections in Kazakh sheep populations. This entailed analyzing the infection correlation between GINs and coccidia and the impact of seasonal climate on GIN and coccidia infections. An epidemiological analysis of GIN and coccidia infections in Kazakh sheep, along with the F1 and F2 generations of Kazakh × Texel crossed sheep in different seasons is presented here providing phenotypic data for future resistance breeding studies.

## Materials and methods

### Sample collection

All animals utilized in this research were prospectively approved and granted a formal waiver of ethics approval by the Animal Welfare Committee of Shihezi University (Xinjiang, China) with the ethical code: A2019-159-01. Rectal fecal sample collection was done on the following animals: the Kazakh ewe sheep (adult, 4–5 years old, Fig 1), the F1 ewe sheep (F1 generation of Kazakh ewe × Texel ram, 3–4 years old), and the F2 ewe sheep (F2 generation of F1 ewe × Texel ram, 2 years old) in April (spring), July (summer), September (autumn), and December

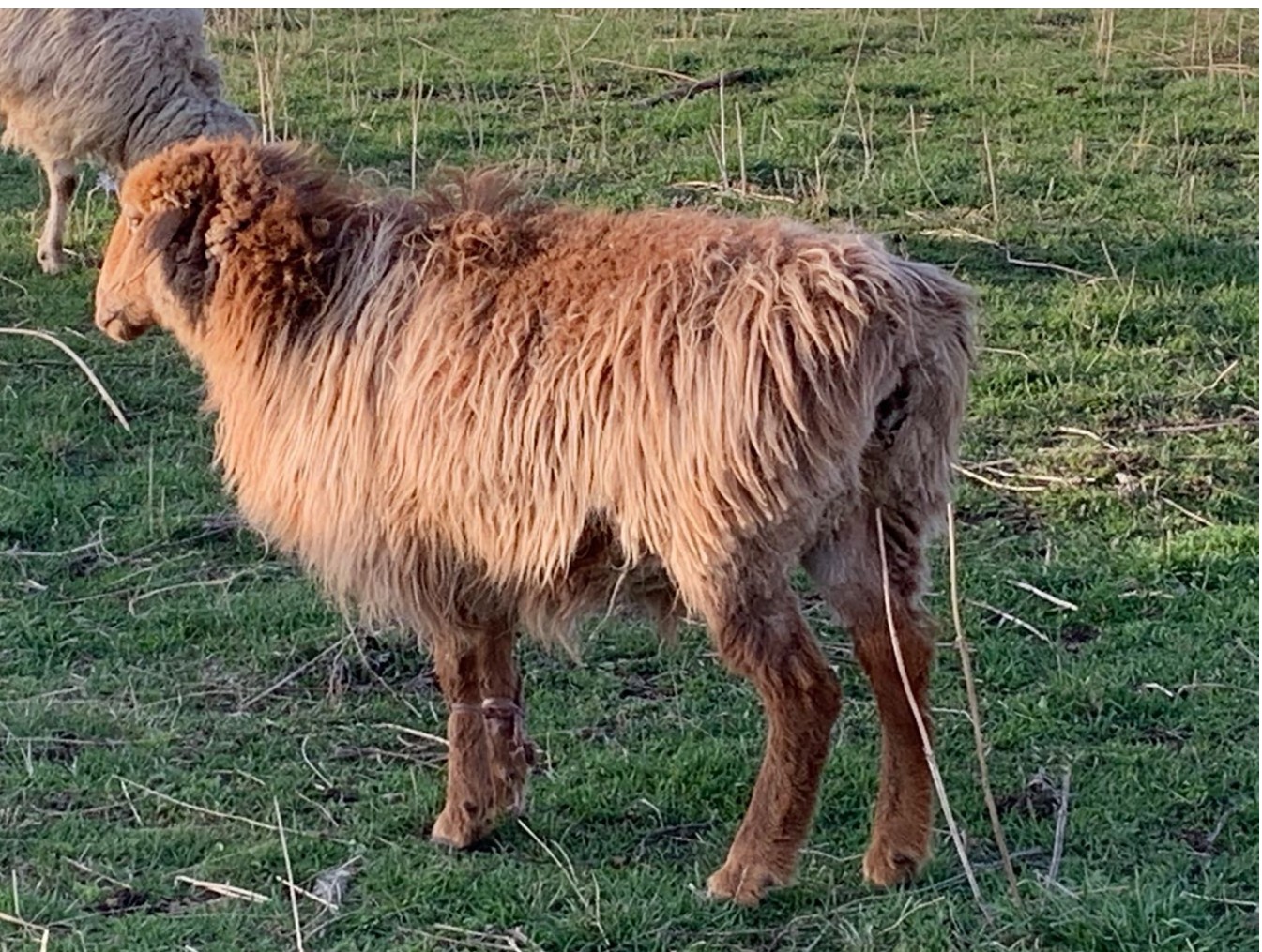

**Fig 1. Kazakh ewe sheep.**

(winter) of 2019 in the Zhaosu County (42˚38'-43˚15' N, 80˚10'-80˚30' E) and the Nilka County (43˚25'-44˚17' N, 81˚85'-84˚58' E) in Ili Kazakh Autonomous Prefecture. The net sample count was 7599 with the details shown in Table 1. Each fecal sample weighed 20–30 g that was sealed in a clean bag and transported to the laboratory and stored at 4˚C. The sheep data such as breed, gender, age, unique chip number, and clinical manifestations were recorded. Deworming was performed following sampling in spring and autumn: with an intramuscular injection of Ivermectin (0.04 ml/kg) and Closantel sodium (0.1–0.2 ml/kg) in spring, and an intramuscular injection of Ivermectin (0.04 ml/kg) and oral Albendazole (0.1–0.15 ml/kg) in autumn. The interval between sampling and deworming was three months.

## Reagents and instruments

The optical microscope utilized was Motic SK200. The digital microscope, microscopic image acquisition and analysis system was Nikon Ci-L. The electronic balance was from Mettler-Toledo Instruments Co., Ltd., model AL104 (d = 0.0001). The modified McMaster egg counting chamber was from the Shanghai Institute of Veterinary Medicine, Chinese Academy of Agricultural Sciences. Salt was purchased from Xinjiang Yanhu Salt Industry Co., Ltd.

## Experimental methods

**Identification of infection types.** This entailed the use of the egg/oocyst floating method. Saturated NaCl was used as the floating fluid (specific gravity 1.2 kg/m$^3$) to check the infecting species in the feces samples. The cover slip was then subjected to microscopic examination. The digital microscope and microscopic image acquisition and analysis system were used to observe the morphology, structure, color, and size of the samples. Subsequently, the images of eggs and oocysts were captured and saved. The species identification was conducted in accordance with reference literature.

**Parasite egg counting.** The eggs of the parasites were counted using the modified McMaster method [31]. The FEC and the FOC in these two chambers were counted under a microscope. The average FEC and FOC values were then multiplied by 200 to obtain the eggs per gram of feces (EPG) and oocysts per gram of feces (OPG) values, respectively.

## Data calculation and statistical analysis

The infection rate and infection intensity were calculated using the following expressions: Infection rate = (number of infected samples / total number of samples) × 100%; The infection intensity included minimum, maximum and average infection intensity. Mean infection intensity = (sum of infection intensity of infected samples / total number of infected samples) ± standard deviation.

SPSS19.0 was used to perform log (n+10) transformation of the EPG and OPG values of a single sample. Normal distribution was observed in the FEC and FOC data. Then the average infection intensity was calculated. The independent sample t-test and one-way analysis of variance were performed to test the differences between seasons, regions, populations and parasite species, with a confidence interval of 95%.

**Table 1. Sample collection in Zhaosu and Nilka counties in the four seasons (number of samples).**

| Sampling Site | Spring (Kazakh/F1) | Summer (Kazakh/ F1/F2) | Autumn (Kazakh/F1) | Winter (Kazakh/F1) |
|---|---|---|---|---|
| Zhaosu | 773/812 | 589/552/262 | 544/743 | 574/803 |
| Nilka | 442 | 459 | 661 | 385 |
| Overall | 2027 | 1862 | 1948 | 1762 |

## Results

### Sheep GIN and coccidia infections

From the fecal examination results, it can be seen that the GIN infection rate was very high. 96.96% of adult sheep were infected by at least one type of GIN at an average infection intensity of 1274.92 ± 2123.21 eggs/gram; the highest value being 42300 eggs/gram. The GIN infection could be generally divided into mild infection (FEC < 500 eggs/gram), moderate infection (FEC = 500–1000 eggs/gram), and severe infection (FEC > 1500 eggs/gram) [32]. The GIN infection in this sample set was found to be mild to moderate (Fig 2), with only 41 samples showing a whole FEC of over 10000 eggs/gram. The coccidial infection rate was very high too at 90.89%. The average infection intensity was 785.11 ± 2385.73 oocysts/gram: the highest being 144500 oocysts/gram. The coccidia infection is divided into the following grades: no infection (OPG = 0 oocysts/gram), mild infection (OPG < 1800 oocysts/gram), moderate infection (OPG = 1800–6000 oocysts/gram), severe infection (OPG > 6000 oocysts/gram) [10]. Most of the coccidia infection in our sample set was mild to moderate (Fig 3). There were 21 samples with the FOC greater than 10000 oocysts/gram.

### Sheep GIN and coccidia infections in different regions

The sheep infection rates for GIN and coccidia in the Zhaosu and Nilka areas were different. While the sheep GIN infection rate in Zhaosu was lower than that observed in Nilka the average sheep GIN infection intensity in Zhaosu was 1386.21 ± 2350.90 eggs/gram, higher than that of Nilka that reported a value of 796.30 ± 968.98 eggs/gram. The mean log (FEC) value of Zhaosu was significantly higher than that of Nilka ($P < 0.001$, Table 2). The sheep coccidia infection rate of Zhaosu was lower than that of Nilka too with the average sheep coccidia infection intensity at 548.81 ± 757.67 oocysts/gram vs. 1190.96 ± 4292.51 oocysts/gram for Nilka. The average log (FOC) value of Zhaosu was also significantly lower than that of Nilka ($P < 0.001$, Table 2).

## Number of samples

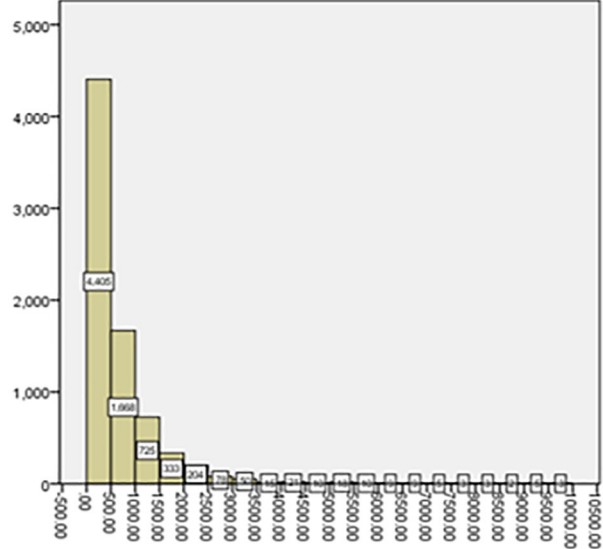

**Fig 2. Distribution of sheep GIN infection intensity.**

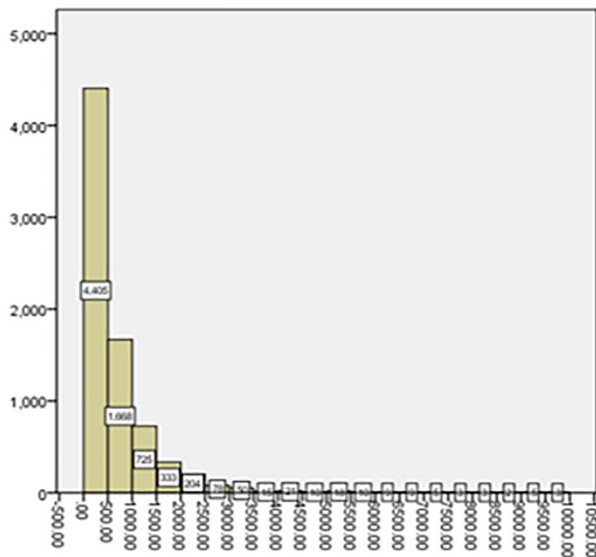

**Fig 3. Distribution of sheep coccidia infection intensity.**

## Sheep GIN and coccidia infections in different regions and different seasons

Regarding the four seasons, the overall infection rate of sheep GIN was the highest in the winter, followed by summer and spring, with the lowest in autumn. In Zhaosu, the highest GIN infection occurred in spring, whereas in Nilka the highest GIN infection occurred in autumn (Table 3). The overall mean log (FEC) of sheep GIN was the highest in spring, followed by summer, then autumn with the lowest in winter ($P < 0.001$, Table 5). For the Zhaosu County, the mean log (FEC) of GIN infection was the highest in spring, significantly higher than that in summer, autumn and winter ($P < 0.001$), while no difference was seen between autumn and winter ($P > 0.001$). For the Nilka County, the mean log(FEC) for GIN infection was the highest in summer ($P < 0.001$), when compared to spring, autumn and winter ($P < 0.001$). The mean log (FEC) in autumn was significantly higher than in spring ($P < 0.01$), which in turn was then significantly more than that in winter ($P < 0.001$, Table 3).

Regarding the four seasons, the overall infection rate of sheep coccidia was the highest in winter, followed by in spring and autumn with the lowest in summer. It was the highest in winter for both Zhaosu and Nilka counties (Table 5). The overall mean log (FOC) of sheep coccidia was the highest in spring, followed by autumn, and the lowest was observed in

**Table 2. Sheep GIN and coccidia infection in different regions.**

| Sampling Site | GIN | | | Coccidia | | |
|---|---|---|---|---|---|---|
| | Infection Rate (%) | Infection Range (eggs/gram) | Mean log(FEC) | Infection Rate (%) | Infection Range (oocysts/gram) | Mean log(FOC) |
| Zhaosu | 96.85% | 0–42,300 | 2.75 [A] | 88.00% | 0–19,000 | 2.40 [A] |
| Nilka | 97.33% | 0–23,950 | 2.68 [B] | 97.23% | 0–144,500 | 2.71 [B] |

Data in the same column, without the same uppercase superscripts (A, B) indicate a highly significant difference ($P < 0.001$).

**Table 3. Sheep GIN infection rate (%) and Log (FEC) of different regions for various seasons.**

| Sampling Site | Spring | | | Summer | | | Autumn | | | Winter | | |
|---|---|---|---|---|---|---|---|---|---|---|---|---|
| | Infection Rate (%) | Max log (FEC) | Mean log (FEC) | Infection Rate (%) | Max log (FEC) | Mean log (FEC) | Infection Rate (%) | Max log (FEC) | Mean log (FEC) | Infection Rate (%) | Max log (FEC) | Mean log (FEC) |
| Zhaosu | 98.80 | 4.63 | 3.38 [A] | 96.36 | 3.75 | 2.64 [B] | 93.40 | 3.79 | 2.58 [C] | 98.33 | 3.55 | 2.55 [C] |
| Nilka | 90.27 | 3.89 | 2.81 [A] | 99.13 | 4.38 | 2.90 [B] | 99.39 | 3.88 | 2.80 [C] | 99.48 | 3.08 | 2.33 [D] |
| Overall | 96.94 | 4.63 | 3.26 [A] | 97.05 | 4.38 | 2.71 [B] | 95.43 | 3.88 | 2.66 [C] | 98.58 | 3.55 | 2.50 [D] |

Data in the same line, without the same uppercase superscripts (A–D) indicate a highly significant difference (P < 0.001).

summer and winter (*P* < 0.01) (Table 4). The mean log (FOC) of sheep coccidia in Zhaosu was the highest in spring, significantly higher than that seen in summer, autumn and winter (*P* < 0.01). The mean log (FOC) of sheep coccidia in Nilka was the highest in spring and summer (*P* < 0.01), followed by autumn (*P* < 0.01). It was higher in spring than in autumn (*P* < 0.05), and the lowest was seen in winter (*P* < 0.01, Table 4).

## GIN and coccidia infections in different sheep populations and different seasons

**GIN and coccidia infections in different sheep populations.** The infection rate of GIN in Kazakh sheep was 97.92% while the GIN infection rate in the F1 generation sheep was 95.74% and in F2 generation sheep of summer, the rate was 94.27%. The mean log (FEC) of GIN in Kazakh sheep was 2.83 ± 0.47 that was significantly more than the F1 (2.78 ± 0.55) and the summer F2 (2.15 ± 0.46) (*P* < 0.001). The mean log (FEC) of the F1 was significantly more compared to the summer F2 (*P* < 0.001).

The infection rate of coccidia in Kazakh sheep was 92.75% while for the F1 generation sheep, this rate was 87.25% and 100.00% in the F2 generation sheep of summer. The mean log (FOC) of coccidia in Kazakh sheep was 2.58 ± 0.62 that was significantly higher than the F1 (2.36 ± 0.66) and the summer F2 (2.20 ± 0.36) (*P* < 0.001). The mean log (FOC) of the F1 was significantly higher than the summer F2 (*P* < 0.001).

**GIN infection in different sheep populations in various seasons.** The GIN infection rate in F1 generation sheep was higher than that of Kazakh sheep only in spring. In the other seasons, the infection rate was higher in Kazakh sheep vs. the F1 and F2 generations. The GIN infection rate and mean log (FEC) of the F1 generation sheep was significantly higher in comparison with the Kazakh sheep (*P* < 0.01) in spring while in summer, the Kazakh sheep displayed significantly higher values than the F1 and F2 generations. In autumn, the values for, Kazakh sheep were significantly higher than the F1 generation (*P* < 0.01). In winter, the

**Table 4. Sheep coccidia infection rate (%) and Log (FOC) of different regions for various seasons.**

| Sampling Site | Spring | | | Summer | | | Autumn | | | Winter | | |
|---|---|---|---|---|---|---|---|---|---|---|---|---|
| | Infecti-on Rate (%) | Max log (FOC) | Mean log (FOC) | Infection Rate (%) | Max log (FOC) | Mean log (FOC) | Infecti-on Rate (%) | Max log (FOC) | Mean log (FOC) | Infection Rate (%) | Max log (FOC) | Mean log (FOC) |
| Zhaosu | 91.42 | 4.04 | 2.76 [A] | 73.91 | 4.10 | 2.10 [B] | 88.27 | 3.52 | 2.37 [C] | 98.18 | 4.28 | 2.32 [C] |
| Nilka | 90.50 | 5.16 | 2.81 [Aa] | 99.13 | 4.76 | 2.93 [A] | 98.79 | 4.00 | 2.70 [Bb] | 100.00 | 3.59 | 2.34 [C] |
| Overall | 91.22 | 5.16 | 2.77 [A] | 80.13 | 4.76 | 2.30 [C] | 91.84 | 4.00 | 2.48 [B] | 98.58 | 4.28 | 2.33 [C] |

Data in the same line, without the same uppercase superscripts (A–D) indicate a highly significant difference (P < 0.01), without the same lowercase superscripts (a–d) differ significantly (P < 0.05).

**Table 5. GIN infection rate (%) and log (FEC) of different sheep populations for various seasons.**

| Populati-on | Spring | | | Summer | | | Autumn | | | Winter | | |
|---|---|---|---|---|---|---|---|---|---|---|---|---|
| | Infecti-on Rate (%) | Max log (FEC) | Mean log (FEC) | Infection Rate (%) | Max log (FEC) | Mean log (FEC) | Infection Rate (%) | Max log (FEC) | Mean log (FEC) | InfectionRate (%) | Max log (FEC) | Mean log (FEC) |
| Kazakh | 95.64 | 4.53 | 3.07$^A$ | 98.09 | 4.38 | 2.80 $^A$ | 99.09 | 3.88 | 2.73 $^A$ | 99.17 | 3.55 | 2.50 $^A$ |
| F1 | 98.89 | 4.63 | 3.36 $^B$ | 96.38 | 3.69 | 2.60 $^B$ | 89.50 | 3.56 | 2.34 $^B$ | 97.88 | 3.28 | 2.42 $^A$ |
| F2 | | | | 94.27 | 3.28 | 2.15 $^C$ | | | | | | |

Data in the same column, without the same uppercase superscripts (A-C) indicate a highly significant difference ($P < 0.01$).

Kazakh and F1 generation sheep showed no differences in GIN infection rate or mean log (FEC) ($P > 0.05$, Table 5).

**Coccidia infection in different sheep populations in various seasons.** The rate of coccidia infection among the Kazakh sheep was higher than that of the F1 generation in summer, autumn, and winter. In spring, the rate of coccidia infection was slightly higher in the F1 generation compared to the Kazakh sheep. In summer, the rate of coccidia infection in the F2 generation was higher than that observed in the F1 generation and the Kazakh sheep (Table 6). The mean log (FOC) of oocysts among the Kazakh sheep was extremely high compared to that of the F1 and F2 generations in spring, summer and autumn ($P < 0.01$). Moreover, the F2 generation displayed an evidently larger value compared to the F1 generation in summer ($P < 0.01$). In winter, there was no significant difference in the mean log (FOC) of oocysts between the Kazakh sheep and the F1 generation ($P > 0.05$, Table 6).

**Correlation analysis of GIN and coccidia infections in different sheep populations.** The FEC elevated with increase in the severity of Eimeria coccidia infection in general. The infection intensity of coccidia had a highly significant effect on the infection intensity of nematodes ($P < 0.001$, Fig 4-1), exhibiting a positive correlation coefficient of 0.187 ($P < 0.001$). The infection intensity of coccidia in Kazakh sheep demonstrated a highly significant effect on the infection intensity of nematodes ($P < 0.001$, Fig 4-2), with a positive correlation coefficient of 0.124 ($P < 0.001$). The infection intensity of coccidia in the F1 generation also displayed a highly significant effect on the nematode infection intensity ($P < 0.001$, Fig 4-3) with a positive correlation coefficient of 0.223 ($P < 0.001$). However, there was no correlation (value = -0.035) between the infection intensity of coccidia and nematode in the F2 generation ($P > 0.05$, Fig 4-4).

## Identification GIN species in sheep

**GIN egg identification.** On the basis of the morphological characteristics of eggs and referral to "Veterinary Parasitology" (Second Edition) [33] and "Veterinary Clinical Parasitology" (Eighth Edition) [34], a total of nine species of gastrointestinal nematodes were initially identified (Fig 5).

**Table 6. Coccidia infection rate (%) and log(FOC) of different sheep populations for various seasons.**

| Populati-on | Spring | | | Summer | | | Autumn | | | Winter | | |
|---|---|---|---|---|---|---|---|---|---|---|---|---|
| | Infecti-on Rate (%) | Max log (FOC) | Mean log (FOC) | Infecti-on Rate (%) | Max log (FOC) | Mean log (FOC) | Infection Rate (%) | Max log (FOC) | Mean log (FOC) | InfectionRate (%) | Max log (FOC) | Mean log (FOC) |
| Kazakh | 91.11 | 5.16 | 2.79$^A$ | 87.02 | 4.76 | 2.52 $^A$ | 94.37 | 3.88 | 2.58 $^A$ | 99.06 | 4.28 | 2.36 $^A$ |
| F1 | 91.38 | 3.79 | 2.75 $^B$ | 64.86 | 3.38 | 1.93 $^B$ | 87.75 | 4.00 | 2.33 $^B$ | 98.01 | 3.58 | 2.29 $^A$ |
| F2 | | | | 100 | 3.65 | 2.20 $^C$ | | | | | | |

Same as Table 5.

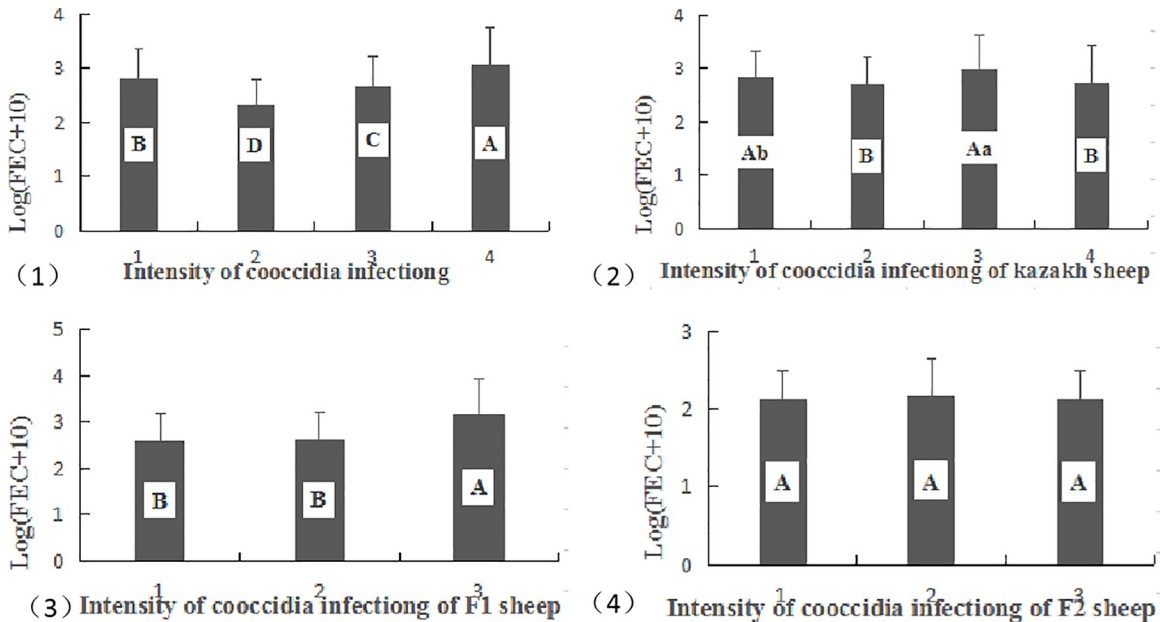

**Fig 4. Correlation between GIN and coccidia infections in different sheep populations.** Data in the same line, without the same uppercase superscripts (A–C) indicate a highly significant difference (*P < 0.001*), without the same lowercase superscripts (a–c) differ significantly (*P < 0.05*).

**Dominant GIN species in different sheep populations in various seasons.** The dominant GIN species in Kazakh sheep included *H. contortus* (90.06%), *Trichostrongylus* spp. (68.53%), and *Ostertagia* spp. (48.86%). For F1 sheep, these were inclusive of *H. contortus* (85.57%), *Trichostrongylus* spp. (59.48%), and *Ostertagia* spp. (33.20%), and for F2 generation the organisms were *H. contortus* (90.08%), *Trichostrongylus* spp. (18.70%), and *B. trigonocephalum* (12.98%). The dominant GIN species for Kazakh and F1 generation sheep in the four seasons were *H. contortus*, *Trichostrongylus* spp., and *Ostertagia* spp., both with a high infection rate and strong infection intensity. But, the infection rate and infection intensity of *B. trigonocephalum* were higher than those of *Ostertagia* spp. in summer (Tables 7–10).

## Discussion

GIN and *Eimeria* coccidia infections in sheep are globally present however with differences in the infection rate and infection intensity attributed to variations in sheep's immunity, climate, feeding and management conditions. This study conducted an investigation to reveal a GIN infection rate of 96.96% at an average infection intensity of 1274.92 ± 2123.21 eggs/gram with the highest value at 42300. Our results are on the same lines of a previous study on the seasonal dynamics of parasites of sheep in Zhaosu that showed a GIN infection rate of 100% at an average intensity of 1967 eggs/gram at a range of 12–12027 [35]. However, the infection rate was higher than other reported areas in Xinjiang, such as an infection rate of 72.34% in the Shihezi area [36], an infection rate of 54.10% in Fukang City [37], and an infection rate of 36.53% in Urumqi County [9].

The average infection rate of sheep GIN in eastern Inner Mongolia was reported at 79.2% with the highest infection rate being 100% that is consistent with the results of this study. However, the average infection intensity was 1813.2 that is higher than what we report [38]. The infection rate and infection intensity of this study were both higher than studies conducted

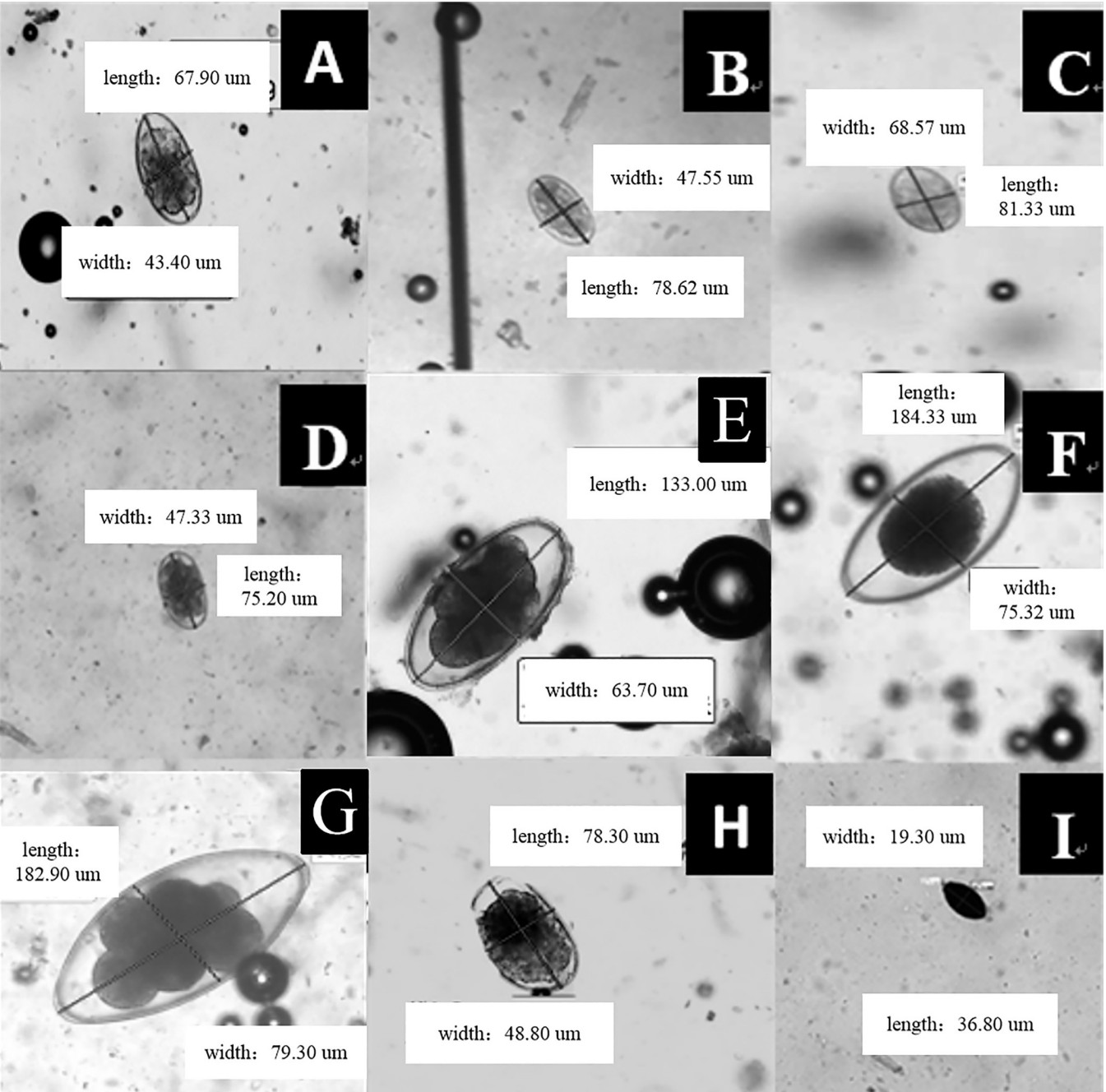

**Fig 5. Identification of GINs in sheep.** A. *Trichostrongylus* spp.; B. *Haemonchus contortus*; C. *Ostertagia* spp.; D. *Bunostomum trigonocephalum*; E. *Chabertia* spp.; F. *Marshallagia* spp.; G. *Nematodirus* spp.; H. *Oesophagostomum* spp.; and I. *Trichuris globulosa*.

elsewhere. Idris A et al [10] reported an infection rate of 62.8% in Germany with an average infection intensity of 315.3 ± 776.8 at a range of 50–17000. Acevedo-Ramírez et al [39] reported that the GIN infection rate of Dorset ewes in the mountains of central Mexico was 44.6%, with a detected FEC range of 0–2450 ± 1137, while the infection rate of Suffolk ewes was 55%, with an FEC range 50–12684.6 ± 4064.2. Although the infection rate of GIN in this study was generally higher than that in other regions, the infection intensity was lower i.e.

**Table 7. Dominant species of sheep GINs in spring.**

| Population | Parasite Species | Infection Rate | Max Infection intensity | Mean infection intensity |
|---|---|---|---|---|
| Kazakh | *H. contortus* | 87.49% | 26700 | 1028.62 |
| | *Trichostrongylus* spp. | 70.45% | 7600 | 696.96 |
| | *Ostertagia* spp. | 62.63% | 6300 | 839.16 |
| F1 | *H. contortus* | 84.11% | 28900 | 1534.55 |
| | *Trichostrongylus* spp. | 76.72% | 5400 | 1070.35 |
| | *Ostertagia* spp. | 73.89% | 10300 | 1104.50 |

mild to moderate infections. In this study, the coccidia infection rate was 90.89%, that is close to the overall infection rate of 92.80% in the three areas of Hebei Province including Baoding, Langfang and Handan [40] and the overall infection rate of 94.80% in Henan, Shandong and Northeast [41]. However, it was higher than some areas in Xinjiang, such as 39.60% in Fukang City [37], 50.90% in Hotan [42], and 71.38% in Urumqi County [9]. The infection rate was higher than what was reported for South Australia (80.00%) [43], and 66.18% in the semi-arid area of northeastern Brazil [44]. In this study, a total of nine species of GIN eggs and *Eimeria* oocysts coccidia were preliminarily identified based on the morphological characteristics, sizes, and the eggs fullness of GIN eggs and *Eimeria* oocysts. However, thenot accurately identification did not involve hatched L3 larvae and molecular biology methods. Due to the morphological similarity of Strongyloides eggs, the possibility of errors in the species identification cannot be ruled out. Nevertheless, this could still be a preliminary presentation of the species responsible for gastrointestinal parasite infections in this region as the dominant species reported here were consistent with most other regions [1, 2, 3, 39]. The GIN and coccidia infection rates revealed in this study were relatively high. The subjects of the study were adult ewes with hardly any sheep showing any obvious clinical signs as most infections were chronic or recessive. Therefore, as a normal control approach firstly, strict deworming must be carried out and secondly, the ewes should be separated from the lambs to avoid the occurrence of serious infections in the latter.

In this study, a positive correlation was found between the EPG and OPG levels in sheep that is consistent with the results reported by Kanyari PW [6] and Idris et al [10]. However, this is in contrast with a report by Kanyari [45] in goats in Queensland (Australia) that recorded a negative correlation. The reason for this discrepancy is that, in Australia (as opposed to Kenya), farmers regularly treat their goats to control helminths but not coccidian parasites hence creating an artificial numerical relationship between the two varieties of parasites. Since the mode of infection is usually per os and fecal contamination of the pastures or

**Table 8. Dominant species of sheep GINs in summer.**

| Population | Parasite Species | Infection Rate | Max Infection intensity | Mean infection intensity |
|---|---|---|---|---|
| Kazakh | *H. contortus* | 90.65% | 5950 | 470.11 |
| | *Trichostrongylus* spp. | 71.37% | 1800 | 272.26 |
| | *B.trigonocephalum* | 52.29% | 11550 | 187.04 |
| F1 | *H. contortus* | 86.41% | 2500 | 296.65 |
| | *Trichostrongylus* spp. | 61.41% | 1600 | 170.80 |
| | *B.trigonocephalum* | 40.04% | 1100 | 132.58 |
| F2 | *H. contortus* | 90.08% | 1650 | 200.85 |
| | *Trichostrongylus* spp. | 18.70% | 200 | 84.69 |
| | *B.trigonocephalum* | 12.98% | 300 | 85.29 |

**Table 9. Dominant species of sheep GINs in autumn.**

| Population | Parasite Species | Infection Rate | Max Infection intensity | Mean infection intensity |
|---|---|---|---|---|
| Kazakh | *H. contortus* | 93.69 | 3700 | 466.47 |
| | *Trichostrongylus* spp. | 69.54% | 2500 | 242.66 |
| | *Ostertagia* spp. | 46.97% | 1600 | 144.52 |
| F1 | *H. contortus* | 83.44% | 2300 | 320.73 |
| | *Trichostrongylus* spp. | 32.97% | 1400 | 178.57 |
| | *Ostertagia* spp. | 19.52% | 1500 | 152.41 |

feed enhances infection of livestock, a positive correlation would be expected in the absence of man's interference by anti- parasitic treatment. Studies have shown that when coccidia were present in animals, there was an increase in nematode eggs too. A test for *H. contortus* revealed that goats that were infected with *Eimeria* (count = 500,000) 60 days earliernot only showed coccidian oocysts, but also more numbers of nematode eggs that was associated withsignificant growth retardation [46]. However, in this study, there was no correlation between F2 sheep GIN and the intensity of coccidiosis infection, which may be due to the small sample size of the F2 that is a limitation here.

As reported, climate change may lead to alterations in parasite epidemiology and infection intensity with regional climate differences exerting a large impact on the epidemiology of GIN infection and its geographic distribution in sheep [47]. The free-living sheep GINs are strongly affected by the climate. Extremely high and low temperatures are harmful to their development and survival. While the increase in temperature usually accelerates their development, the mortality rate too is boosted. The development and transformation of larvae from manure to pasture requires water, so rain is the limiting factor for infection. These factors together influence the seasonal pattern of sheep infections that exhibit the relative importance of geographic differences to the epidemiology of different GIN species in Europe as well [48].

One sampling site of this study was the Zhaosu area, which is located in the northwest of Xinjiang Uygur Autonomous Region and the southwest of Ili Kazakh Autonomous Prefecture. Zhaosu Basin in a high elevation intermountain basin situated in inland Central Asia, and has a semi-arid, semi-humid and cold, temperate continental mountainous climate. It has long winters and short summers, with an annual average temperature of 3.23°C and an annual average precipitation of 507.88 mm [49]. The other sampling site, Nilka area, has a climate similar to Zhaosu, but with an annual average temperature of 8.50°C, higher compared to Zhaosu with an annual average precipitation of 335 mm that is lower than that of Zhaosu [50]. The differences in latitude, temperature, and precipitation in these two regions resulted highly significant differences in the infection intensities of GIN and coccidiosis. The infection intensity of GIN in Zhaosu was significantly higher compared to that in Nilka, whereas the coccidia infection intensity in Nilka was significantly higher than that in Zhaosu. Both these regions have a

**Table 10. Dominant species of sheep GINs in winter.**

| Population | Parasite Species | Infection Rate | Max Infection intensity | Mean infection intensity |
|---|---|---|---|---|
| Kazakh | *H. contortus* | 84.36% | 1950 | 265.80 |
| | *Trichostrongylus* spp. | 61.73% | 1800 | 190.37 |
| | *Ostertagia* spp. | 33.26% | 850 | 106.90 |
| F1 | *H. contortus* | 88.41% | 1400 | 222.18 |
| | *Trichostrongylus* spp. | 65.26% | 650 | 144.85 |
| | *Ostertagia* spp. | 25.53% | 550 | 88.78 |

north temperate continental climate that has significant temperature changes in four seasons. In spring, the temperature varies greatly while the autumn is short and a huge temperature variation in winter, displaying a climate characteristic of distinct four seasons [51].

In this study, the intensity of GIN infection in sheep was the highest in spring, followed by summer, autumn, and was the lowest in winter. For coccidia infection, the infection was the highest in spring, followed by autumn, and the lowest in summer and winter. These results were also closely related to the climate characteristics of these two regions. They were also consistent with the conclusion of a previous study on the seasonal dynamics of sheep GIN infection in the north that "it was significant in spring, followed by in autumn, then in summer, and lowest in winter" [52]. Hutchinson [53] reported that cold stimulus is responsible for arrested development of larvae. During winter, animals are also partially stall-fed that reduces the chance of infection. The period of grazing is also reduced during winter. These along with hypobiosis of pre-parasitic stages all contribute to the low infection during this period. Further, the majority of ewes are pregnant during this period. The hormonal impact results in low fecal egg output and contributes to low infection in pastures in winters.

The GIN infection differences in various populations of sheep are not only affected by age, nutritional levels, health status, and feeding management, but also impacted by genetic differences in resistance towards parasites. Studies have found that sheep breeds originating from areas with high humidity, high temperature and high incidence of parasitic diseases such as Barbados Blackbelly, St. Croix, Red Maasai and Florida have stronger parasite resistance [23, 54]. In an environment with high parasite infections with no anti-helminthic application, the resistant individuals survived and reproduced to develop a strong parasite resistance after long-term natural selection. Therefore, generally speaking, the sheep breeds developed under a continuous and strong environment of disease infection have stronger disease resistance and tolerance compared to the introduced breeds [55].

The successful selection of genetically resistant sheep is associated with the markers used, and FEC is the most popular and feasible method for selecting parasite-resistant sheep [56]. The heritability of small ruminants against GIN measured by the FEC method ranged between 0.01 and 0.65 [57]. At two research centers in New Zealand, the Wallaceville Animal Research Center and the Ruakura Agricultural Research Center, selection lines have already been established for Romney sheep with GIN resistance (low FEC) and GIN susceptibility (high FEC). At log (FEC) values of 0.86 and 1.65, respectively, the FEC was closely related to GIN infection [25]. Idris et al. demonstrated the effect of birth type, gender and breed on GIN infection in sheep [10]. Lambs born from multiple-birth pregnancies had a higher FEC than lambs born from single pregnancies, while male lambs were more susceptible than female lambs. While the FEC difference between breeds was not significant, the difference within breeds was higher, indicating the possibility of selecting parasite resistance in these breeds. Research by Good et al [56] showed a significant difference between Texel and Suffolk sheep in the resistance to GIN in a temperate environment. Regardless of age, Texel had stronger resistance to GIN infection compared to the co-grazed Suffolk. Such stronger resistance may be innate, or because they could produce a faster immune response. The Th2 immune response of GIN-resistant animals is a typical immune response caused by helminth infection. The Th2 response of the Texel sheep was faster and stronger because it was better at recognizing parasite-specific antigens [58]. The F1 and F2 generations of the sheep populations in this study were crosses between Texel and Kazakh sheep, and the infection intensities of both GIN and coccidia were significantly lower than the Kazakh sheep. This is consistent with the GIN resistance displayed by the Texel sheep in the above mentioned studies. This may be attributed to the vital involvement of the Texel breed as the terminal paternal line and in the genetic composition of the breeding ewe population.

## Conclusions

In this work, we studied four different populations of Kazakh sheep in the Zhaosu County and Nilka County in Ili Kazakh Autonomous Prefecture. The epidemiological features of GIN and coccidian infections in these four populations were analyzed across the four seasons. The rates of GIN and coccidian infection were relatively high, and the infection was of mild to moderate intensity in both counties. Zhaosu County and Nilka County are located at different latitudes, and the temperature and humidity also vary. While the intensity of GIN infection was significantly higher in Zhaosu County compared to that of Nilka County the intensity of coccidian infection was apparently lower in the former vs. the latter. Among the four seasons, the intensity of GIN infection was the highest in the two counties in spring, followed by summer and autumn, while it was the lowest in winter. The intensity of coccidian infection was the highest in spring, followed by autumn, and was the lowest in summer and winter. The F1 generation of the Kazakh × Texel cross showed extremely lower rates of GIN and coccidian infection compared to the Kazakh sheep. The present study provides a basis for the screening of candidate genes that are responsible for resistance to GIN infection.

## Author Contributions

**Conceptualization:** Xiaofei Yan.

**Data curation:** Xiaofei Yan.

**Formal analysis:** Xiaofei Yan.

**Investigation:** Xiaofei Yan.

**Methodology:** Xiaofei Yan.

**Resources:** Yiyong Liu, Keqi Ding, Haifeng Deng, Peiming Wang.

**Software:** Ting Tong.

**Writing – original draft:** Xiaofei Yan.

**Writing – review & editing:** Xiaofei Yan, Mingjun Liu, Sangang He.

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
