## [Decision Letter · Decision Letter 0]

11 Mar 2021

PONE-D-21-03023

Epidemiological study on gastrointestinal nematode and coccidia infections in different populations of Kazakh sheep

PLOS ONE

Dear Dr. Yan,

Thank you for submitting your manuscript to PLOS ONE. After careful consideration, we feel that it has merit but does not fully meet PLOS ONE’s publication criteria as it currently stands. Therefore, we invite you to submit a revised version of the manuscript that addresses the points raised during the review process.

It was reviewed by two experts in the field, and they have recommended some modifications be made prior to acceptance.

I therefore invite you to make these changes and to write a response to reviewers which will expedite revision upon resubmission.

We look forward to receiving your revised manuscript.

I wish you the best of luck with your modifications.

Hope you are keeping safe and well in these difficult times.

Kind regards,

Simon Clegg, PhD

Academic Editor

PLOS ONE

"This study was financially supported by the National Gm major special project - transgenic sheep new breed breeding project (2016ZX08008001-001-002), Xinjiang Autonomous Region University scientific research project (XJED2020Y049). The funders had no role in study design, data collection and analysis, decision to publish, or preparation of the manuscript."

3. We note you have included a table to which you do not refer in the text of your manuscript. Please ensure that you refer to Table 9 in your text; if accepted, production will need this reference to link the reader to the Table.

4. We note you currently have your Tables (1 to 15) and (Figures 1 to 4) duplicated as supporting information files. Could you please update the submission to make them either Tables and Figures or Supporting Information files, so they are not duplicated. 

Further details with regards to tables and figures can be found here: https://journals.plos.org/plosone/s/submission-guidelines#loc-figures-and-tables and supporting information can be found here: https://journals.plos.org/plosone/s/submission-guidelines#loc-supporting-information

Reviewers' comments:

Reviewer's Responses to Questions

**Comments to the Author**

1. Is the manuscript technically sound, and do the data support the conclusions?

Reviewer #1: Yes

Reviewer #2: Yes

2. Has the statistical analysis been performed appropriately and rigorously? 

Reviewer #1: Yes

Reviewer #2: Yes

3. Have the authors made all data underlying the findings in their manuscript fully available?

Reviewer #1: Yes

Reviewer #2: Yes

4. Is the manuscript presented in an intelligible fashion and written in standard English?

Reviewer #1: Yes

Reviewer #2: No

5. Review Comments to the Author

Reviewer #1: Minor changes in my opinion, reads really well and think it’s a great paper. I also think that the figures are really great too.

However, I think just adding a photo of a Kazakh sheep would add some more context to this paper for readers who don’t know what one may be.

I also have a comment on a small grammar error which is found on line 44 “As the infection of Eimeria coccidia got severer, FEC appeared to increase somewhat.” – severer is not a word.

It would be really interesting in future to compare this data with another breed of sheep too.

Why was the saturated saline method used over other methods/flotation fluids?

Reviewer #2: This study researched an interesting and important issue of ovine endoparasitic infections and multiparasitism by nematodes and coccidia as well as resistance in various sheep breeds. The sample sizes were robust and it was great that all four seasons were studied. However, there were unfortunately several areas of concern:

1. Overall, the writing style was hard to follow and quite repetitive at times, which could be condensed; spelling, grammar, and sentence structure could all be improved. For example:

- "Coccidioides" are a genus of fungi

- Oocysts are often incorrectly called oocytes

- The first word of "highly significantly higher/bigger" can be removed

- Eimeria genus may be clearer than "Eimeria coccidia"

- Grade 1-4 for coccidia seems unnecessary when you can simply describe them as mild to severe as you did for GIN

- To save space and make it easier to read, solely using a percentage is better than following it with the amount of samples in brackets

2. I am unsure about the aims of this paper: are only Eimeria observed or all coccidia? If the former, then why (as all the nematodes are grouped together) and can you make this clearer in the title? Also, how will this study benefit the wider community? There is not much information on coinfection, Eimeria, seasonality/climate change, etc in the introduction.

3. There are many tables, which may be able to be merged or condensed if not moved to supplementary information rather than on the article itself. The superscript significance captions are sometimes hard to understand due to phrasing. Also, data on the sex of the sheep could be included in Table 1? Why is the data for F2 missing in Tables 8-11?

4. There are a lot of statistics (P values, OPG, etc) in your abstract, introduction, and discussion, which often should not have them. Could you describe them without using these?

5. Would using EPG (eggs per gram) be more suitable than FEC? Or alternately, FOC (fecal oocyst count) instead of OPG to keep things consistent?

6. Some of the details in the methods could be condensed.

7. The discussion could benefit by reducing the number of statistical comparisons to other studies and focusing on explaining the results more thoroughly (why they were found, potential implications/benefits, etc) as well as study limitations/improvements, wider applications, and future research. Also, the coordinates should be moved to the methods section.

8. The graphs were all very blurry and hard to read. Also, are standard error/deviation bars not required? Figure 4 should be "length by width".

I hope these suggestions help and look forward to reading the revised version. Thank you!

6. PLOS authors have the option to publish the peer review history of their article (what does this mean?). If published, this will include your full peer review and any attached files.

Reviewer #2: No

---

## [Author Response · Author response to Decision Letter 0]

12 Apr 2021

Response to Editor and Reviewers

1.Response to editors

（1）Please ensure that your manuscript meets PLOS ONE's style requirements, including those for file naming. The PLOS ONE style templates can be found at https://journals.plos.org/plosone/s/file?id=wjVg/PLOSOne_formatting_sample_main_body.pdf and https://journals.plos.org/plosone/s/file?id=ba62/PLOSOne_formatting_sample_title_authors_affiliations.pdf

Response：The manuscript has been modified according to PLOS ONE style requirements.

（2）Thank you for stating the following in the Acknowledgments Section of your manuscript:

"This study was financially supported by the National Gm major special project - transgenic sheep new breed breeding project (2016ZX08008001-001-002), Xinjiang Autonomous Region University scientific research project (XJED2020Y049). The funders had no role in study design, data collection and analysis, decision to publish, or preparation of the manuscript."

response：

a: The funding-related text was removed from the manuscript.

b: It was included in the revised cover letter.

The funding information is provided as follows: 

This study was financially supported by the KeyLab funding of Xinjiang Ugrus Autonomous region"identification of genes with resistance to sheep gastrointestinal nemotode infection by genome-wide association study", and Scientific Research Project of University of Xinjiang Autonomous Region (XJED2020Y049). 

(3)We note you have included a table to which you do not refer in the text of your manuscript. Please ensure that you refer to Table 9 in your text; if accepted, production will need this reference to link the reader to the Table.

Response：Table 9 was included in the manuscript between lines 240 and 242 in the old manuscript.Tables were merged in the new manuscript. 

(4)We note you currently have your Tables (1 to 15) and (Figures 1 to 4) duplicated as supporting information files. Could you please update the submission to make them either Tables and Figures or Supporting Information files, so they are not duplicated.

Response：The tables and figures were provided by "Supporting Information files" in revised manuscript. 

(5) “Place each table in your manuscript file directly after the paragraph in which it is first cited (read order). Do not submit your tables in separate files”.

Response: All the tables were placed after the paragraph.

2.Response to Reviewer Comments

Reviewer #1: Minor changes in my opinion, reads really well and think it’s a great paper. I also think that the figures are really great too.

However, I think just adding a photo of a Kazakh sheep would add some more context to this paper for readers who don’t know what one may be.

response：A photo of Kazakh sheep was added in manuscript.

I also have a comment on a small grammar error which is found on line 44 “As the infection of Eimeria coccidia got severer, FEC appeared to increase somewhat.” – severer is not a word.

Response：Modification by "A positive correlation was found between the EPG and OPG levels in the sheep.” 

Why was the saturated saline method used over other methods/flotation fluids?

Response：The saturated saline method is a classical and widely used method to measure worm eggs. It is convenient, fast and precise, so we used the method in our work.

Reviewer #2

(1)Overall, the writing style was hard to follow and quite repetitive at times, which could be condensed; spelling, grammar, and sentence structure could all be improved. 

For example:

- "Coccidioides" are a genus of fungi

- Oocysts are often incorrectly called oocytes

- The first word of "highly significantly higher/bigger" can be removed

- Eimeria genus may be clearer than "Eimeria coccidia"

- Grade 1-4 for coccidia seems unnecessary when you can simply describe them as mild to severe as you did for GIN

- To save space and make it easier to read, solely using a percentage is better than following it with the amount of samples in brackets

Response：The language problem including the grammar and structure of the manuscript have been modified and marked in the revised manuscript.

(2)I am unsure about the aims of this paper: are only Eimeria observed or all coccidia? If the former, then why (as all the nematodes are grouped together) and can you make this clearer in the title? Also, how will this study benefit the wider community? There is not much information on coinfection, Eimeria, seasonality/climate change, etc in the introduction.

response：I think only Eimeria observed of this paper, title was modified “Epidemiological study on gastrointestinal nematode and Eimeria coccidia infections in different populations of Kazakh sheep”. I have added GIN and Eimeria coccidia coinfection in the introduction of the revised draft and the influence of seasonality on the infection of GIN and Eimeria coccidia.

(3)There are many tables, which may be able to be merged or condensed if not moved to supplementary information rather than on the article itself. The superscript significance captions are sometimes hard to understand due to phrasing. Also, data on the sex of the sheep could be included in Table 1? Why is the data for F2 missing in Tables 8-11?

Response：PLOS ONE required to place each table in manuscript coming after the paragraph in which it is first cited (read order) . So we put the tables in the text by the order they appear in the body. All the samples were collected from ewes, so the sex was omitted. If needed, it could be listed in Table 1. We only collected the F2 data in the summer, so we did not put it in the table. The significant captions were enlarged and highlighted.

(4)There are a lot of statistics (P values, OPG, etc) in your abstract, introduction, and discussion, which often should not have them. Could you describe them without using these?

Response：The P value and OPG in abstract, introduction and discussion were deleted.

(5)Would using EPG (eggs per gram) be more suitable than FEC? Or alternately, FOC (fecal oocyst count) instead of OPG to keep things consistent?

Response：Replace EPG with FEC and OPG with FOC in the full text.

(6)Some of the details in the methods could be condensed.

Response：Part of he methods were modified and made it more concise..

(7)The discussion could benefit by reducing the number of statistical comparisons to other studies and focusing on explaining the results more thoroughly (why they were found, potential implications/benefits, etc) as well as study limitations/improvements, wider applications, and future research. Also, the coordinates should be moved to the methods section

Response：The discussion was modified. More analysis and discussions were added.The coordinates were moved to the method section.

(8)The graphs were all very blurry and hard to read. Also, are standard error/deviation bars not required? Figure 4 should be "length by width"

Response：Some of the standard error were deleted，and Figure 4 was modified as "length" and "width".

---

## [Editor Report · Decision Letter 1]

26 Apr 2021

An epidemiological study of gastrointestinal nematode and Eimeria  coccidia infections in different populations of Kazakh sheep

PONE-D-21-03023R1

Dear Dr. Yan,

We’re pleased to inform you that your manuscript has been judged scientifically suitable for publication and will be formally accepted for publication once it meets all outstanding technical requirements.

Kind regards,

Simon Clegg, PhD

Academic Editor

PLOS ONE

Additional Editor Comments:

Many thanks for resubmitting your manuscript to PLOS One

As you have addressed all the comments and the manuscript reads well, I have recommended it for publication

You should hear from the Editorial Office shortly.

If you could make the following minor modifications during editing I would be most grateful

Line 121- space between at an

Line 125- space between too at

Line 246- eggs written incorrectly

Line 315- space between above and mentioned

It was a pleasure working with you and I wish you the best of luck for your future research

Hope you are keeping safe and well in these difficult times

Thanks

Simon

---

## [Editor Report · Acceptance letter]

10 May 2021

PONE-D-21-03023R1 

An epidemiological study of gastrointestinal nematode and *Eimeria* coccidia infections in different populations of Kazakh sheep 

Dear Dr. Yan:

I'm pleased to inform you that your manuscript has been deemed suitable for publication in PLOS ONE. Congratulations! Your manuscript is now with our production department. 

Kind regards, 

on behalf of

Dr. Simon Clegg 

Academic Editor

PLOS ONE